# Cell-specific expression of key mitochondrial enzymes limits OXPHOS in astrocytes of the adult human neocortex and hippocampal formation
Arpád Dobolyi [1,2], Melinda Cservenák[2], Attila G. Bagó[3], Chun Chen[4], Anna Stepanova[5], Krisztina Paál[6], Jeonghyoun Lee[6], Miklós Palkovits[1,7], Gavin Hudson[4] & Christos Chinopoulos [6] ✉

The astrocyte-to-neuron lactate shuttle model entails that, upon glutamatergic neurotransmission, glycolytically derived pyruvate in astrocytes is mainly converted to lactate instead of being entirely catabolized in mitochondria. The mechanism of this metabolic rewiring and its occurrence in human brain are unclear. Here by using immunohistochemistry (4 brains) and imaging mass cytometry (8 brains) we show that astrocytes of the adult human neocortex and hippocampal formation express barely detectable amounts of mitochondrial proteins critical for performing oxidative phosphorylation (OXPHOS). These data are corroborated by queries of transcriptomes (107 brains) of neuronal versus non-neuronal cells fetched from the Allen Institute for Brain Science for genes coding for a much larger repertoire of entities contributing to OXPHOS, showing that human non-neuronal elements barely expressed mRNAs coding for such proteins. With less OXPHOS, human brain astrocytes are thus bound to produce more lactate to avoid interruption of glycolysis.

The Astrocyte-To-Neuron Lactate Shuttle (ANLS) model dictates that in response to glutamate-mediated neuronal activity, astrocytes enhance their glycolytic flux forming lactate, which is shuttled to neurons through monocarboxylate transporters[1]. An understated but key point of this model is that in astrocytes, glycolytically produced pyruvate is converted to lactate instead of entering mitochondria to get oxidatively catabolized, even in the presence of oxygen[2]. This evolutionary conserved[3] astrocyte-specific "aerobic glycolysis" -more commonly associated with cancer[4]- has never been adequately explained. In the original scheme of the article proposing the ANLS hypothesis, mitochondria were tactfully missing from astrocytes[1], and this has remained unchanged ever since[5,6]. Pellerin and Magistretti attributed the conversion of pyruvate into lactate in astrocytes to a lack of mitochondrial aspartate/glutamate carrier that reduces their capacity to transfer NADH by the malate/aspartate shuttle (MAS) in the mitochondria in order to regenerate $NAD^+$[5]. Thus, lactate

dehydrogenase would be essential for regenerating $NAD^+$ which is indispensable for glyceraldehyde dehydrogenase, allowing glycolysis to proceed[7].

However, the absence of MAS does not universally imply upregulation of glycolysis, see for example[8]. Furthermore, MAS inhibition leads only to an incomplete prevention of mitochondrial utilization of pyruvate[9]. Finally, it was recently discovered that cytosolic malate dehydrogenase (MDH1) may also assume the role of regenerating $NAD^+$ when mitochondria are dysfunctional[10]. The increase in astrocytic glycolytic flux could also be explained via a mechanism involving $Na^+/K^+$ ATPase[1], however, that cannot account for pyruvate not getting metabolized in mitochondria. The groups of Bolaños, Felipe Barros and Clark have offered the explanation that NO -which is formed and released by neurons during glutamatergic neurotransmission[11]- activates glycolysis[12] simultaneously to inhibiting complex IV in astrocytes[13] at nanomolar concentrations[14] by reversibly competing with oxygen[15]. These effects cause the activation of glycolysis in

[1]Laboratory of Neuromorphology, Department of Anatomy, Histology and Embryology, Semmelweis University, Budapest, Hungary. [2]Laboratory of Molecular and Systems Neurobiology, Department of Physiology and Neurobiology, Eotvos Lorand University, Budapest, Hungary. [3]National Institute of Mental Health, Neurology and Neurosurgery, Department of Surgical Neurooncology, Budapest, Hungary. [4]Wellcome Centre for Mitochondrial Research, Bioscience Institute, Newcastle University, Newcastle upon Tyne, UK. [5]Brain and Mind Research Institute, Weill Cornell Medical College, New York, NY, USA. [6]Institute of Biochemistry and Molecular Biology, Department of Biochemistry, Semmelweis University, Budapest, Hungary. [7]Human Brain Tissue Bank, Semmelweis University, Budapest, Hungary. ✉e-mail: chinopoulos.christos@semmelweis.hu

astrocytes releasing lactate[16]. The groups of Bolaños have also reported that murine neuronal mitochondrial complex I is assembled into super-complexes increasing OXPHOS efficacy, whereas the majority of astrocytic complex I is free[17]. In view of this astrocytic OXPHOS inefficiency, there should be pyruvate -> lactate formation in order to regenerate cytosolic NAD$^+$. Finally, the group of Sokoloff published that the activity of rat astrocytic pyruvate dehydrogenase complex is low due to hyperphosphorylation, disfavouring mitochondrial utilization of pyruvate, thus diverting it towards lactate[18].

The consensus that astrocytic energy demands are mostly met by glycolysis, whereas those of neurons mainly rely on OXPHOS is almost universally accepted (see[16] for in-favour and[19] for against); however, it is entirely based on work from cells or brain tissues of laboratory rodents[1,20,21] or induced human stem cells (see under Results and Discussion). On the other hand, the group of McKenna demonstrated that murine astrocytes exhibit the capacity for oxidative decarboxylation of glutamate[22], which is consistent with the finding that mouse astrocytes express components of the citric acid cycle and the electron transport chain in vivo[23,24]. Despite that the ANLS model has been demonstrated in the human brain in vivo[25], Dienel and Yellen question its validity[26,27], mostly on the grounds that murine astrocytes exhibit the capacity for performing the citric acid cycle and oxidative phosphorylation on the basis of transcriptomic, proteomic and functional analysis. However, the group of Chinopoulos showed that subunits coding for succinyl-CoA ligase and the α-ketoglutarate dehydrogenase complex (KGDHC) -two entities of the citric acid cycle- are lacking from adult human brain astrocytes[28,29]. Lack of KGDHC immunoreactivity from human glia has also been reported earlier by Blass[30]. This severely truncated citric acid cycle present in adult human astrocytes, in conjunction with the reports that astrocytic glycolysis yields lactate with little -if any- entry of pyruvate into mitochondria, led us to investigate the cell-specific expression of OXPHOS-critical enzymes.

## Results and Discussion

All antibodies used in this study have been firmly established as valid, monospecific antibodies for the techniques and tissue types used hereby. Proteins required for OXPHOS, cytochrome c and Cox IV were found highly predominantly in neurons in the cerebral cortex and hippocampus based on immunolabelling of large labelled cell bodies and also the morphology of the cell bodies and visible primary dendrites. These characteristics of cytochrome c and Cox IV protein expression were the same in all investigated cortical brain areas including the subiculum (Fig. 1), the dorsomedial prefrontal cortex (Fig. 2), the dentate gyrus (Supplementary Fig. 1 in the Supplementary Information), and the parahippocampal cortex (Supplementary Fig. 2). In turn, in the substantia nigra, a low level of CoxIV immunoreactivity was detected in astrocytes, albeit at a much smaller amount than in neurons (Fig. 3). The presence of labelled cell bodies in grey but not white matter confirmed the mostly neuronal but not astrocytic localization in the human cortical brain areas. Double immunolabelling with the astrocytic marker S100B protein also demonstrated that astrocytes contain Cox IV protein to a very small degree in the subiculum (Fig. 1F) and in the frontal cortex (Fig. 2E, quantified in 2 F). It must be stressed that astrocytes can have long processes, some of which may have escaped detection using the astrocytic marker S100B. However, the imaging mass cytometry data used GFAP (see below and Fig. 4) which stains better than S100B regarding the astrocytic long processes, yet support the same conclusions, namely that astrocytic marker-positive cells harbor OXPHOS-critical enzymes in cortical and hippocampal areas to a lower level than in neurons. The percentage of the cell body covered by CoxIV-immunolabelled particles was $7.82 \pm 0.88$ ($n = 28$ cells) and $9.83 \pm 0.84$ ($n = 52$ cells) for neurons in the two brain areas, respectively. In contrast, these values were $0.32 \pm 0.18$ ($n = 18$ cells) and $1.14 \pm 0.14$ ($n = 52$ cells) for astrocytes. These data are entirely congruent with those reporting Cox IV cell-specific enzymatic activity or immunoreactivity in situ in animals placed in high positions in the phylogenetic tree (see below), including humans suffering from Parkinson's disease[31]; furthermore, the higher the

position of the organism in the phylogenetic tree, and the higher brain region investigated, the less the Cox IV enzymatic activity. However, this was less pertinent in the human substantia nigra, where we observed a considerably higher level of Cox IV immunoreactivity in astrocytes compared to the other brain regions, albeit still less than in neurons (Fig. 3). The percentage of the cell body covered by CoxIV-immunolabelled particles was $37.18 \pm 1.24$ ($n = 64$ cells) for neurons in the substantia nigra while for astrocytes, the value was $6.34 \pm 0.76$ ($n = 59$ cells).

The finding of much less CoxIV in astrocytes than in neurons was confirmed in the CA1 region of the hippocampus using imaging mass cytometry technique[32] (Fig. 4). Using a larger sample size ($n = 8$ brains), the amount of percentage of the cell body covered by CoxIV-immunolabelled particles was $4.97 \pm 0.55$ for neurons ($n = 100$ cells) identified by neural Hu protein (HuB+C + D)[33] labelling while $1.41 \pm 0.18$ for astrocytes ($n = 100$ cells) identified by labelling with glial fibrillary acidic protein (GFAP).

In previous studies Cox IV activity was dominantly but not completely expressed in neurons in auditory relay nuclei of cat as compared to non-neuronal neuropil[34]; in the Lateral Geniculate Nucleus of cat, Cox IV immunoreactivity was present only in the neurons[35]; In monkey hippocampus, Cox IV immunoreactivity was not present in glia[36] (in this reference, the same was also reported for mouse cerebellum); in macaque cerebellum and hippocampus, very low Cox IV reactivity was reported in non-neuronally populated areas[37]; in primate visual cortex of macaque <2% of Cox IV immunoreactivity was present in glia in ref. [38]; in the lateral entorhinal cortex of the rhesus monkey, Cox IV immunoreactivity was detected only in the neurons; In human entorhinal cortex, Cox IV immunoreactivity was absent from glial elements[39]; In ref. [40], no Cox IV enzymatic activity staining was observed in the developing and adult human cortical layers where non-neuronal elements are found. In ref. [41], it was shown that Cox IV mRNA was localized mainly or exclusively in cell bodies and proximal dendrites; this is consistent with our query in the Allen Institute for Brain Science showing that the mRNAs of cytochrome c and Cox IV are expressed in neurons to a much higher extent than in non-neuronal cell types (Fig. 1A and Supplementary Fig. 7). In contrast to the very low or undetectable extent of immunolabelling of astrocytic Cox IV in the human cortical areas, Cox IV was found in both neuronal and astrocytic cell types in the rat cerebral cortex (compare Supplementary Fig. panel 3A, B for rat *vs* 3 C for human). Likewise, we also found that in mouse brain, RNA-Seq data fetched from the Allen Institute for Brain Science suggested that the mRNAs of Cox IV subunits are expressed in neurons and non-neuronal cell types to a similar degree (Supplementary Fig. 8). Heatmaps of mRNA expression levels in neuronal (Neu) and non-neuronal (N-N) cells based on single cell sequencing by the Allen Institute for Brain Science for all subunits participating in OXPHOS in humans and mice is provided in the Supplementary Data 2 (Dataset homo *vs* mus, respectively).

As opposed to cytochrome c and Cox IV, pyruvate carboxylase was found exclusively in astrocytes based on the morphology and localization of the cells (Fig. 5A) as well as double labelling with S100B (Fig. 5G) consistent with literature on mice[42]. The percentage of the cell body covered by pyruvate carboxylase-immunolabelled particles was $13.42 \pm 0.96$ ($n = 36$ cells) in astrocytes while neurons were not detected by them (Supplementary Fig. 4A). Pyruvate transporters were found in both neuronal and astrocytic cell types, but with differential expression as MPC1 was found only in neurons while MPC2 only in astrocytes in the subiculum (Fig. 5B, C), the dentate gyrus (Supplementary Fig. 1) and the parahippocampal gyrus (Supplementary Fig. 2). We confirmed the almost predominant neuronal localization of MPC1 using double labelling with S100B as well (Fig. 5H). The percentage of the cell body covered by MPC1-immunolabelled particles was $14.78 \pm 0.70$ ($n = 56$ cells) in neurons and only $0.98 \pm 0.13$ ($n = 50$ cells) in astrocytes (Supplementary Fig. 4B). Additional pyruvate utilization enzymes are present in both neurons and astrocytes as we found three different subunits of pyruvate dehydrogenase (PDH) localized in both cell

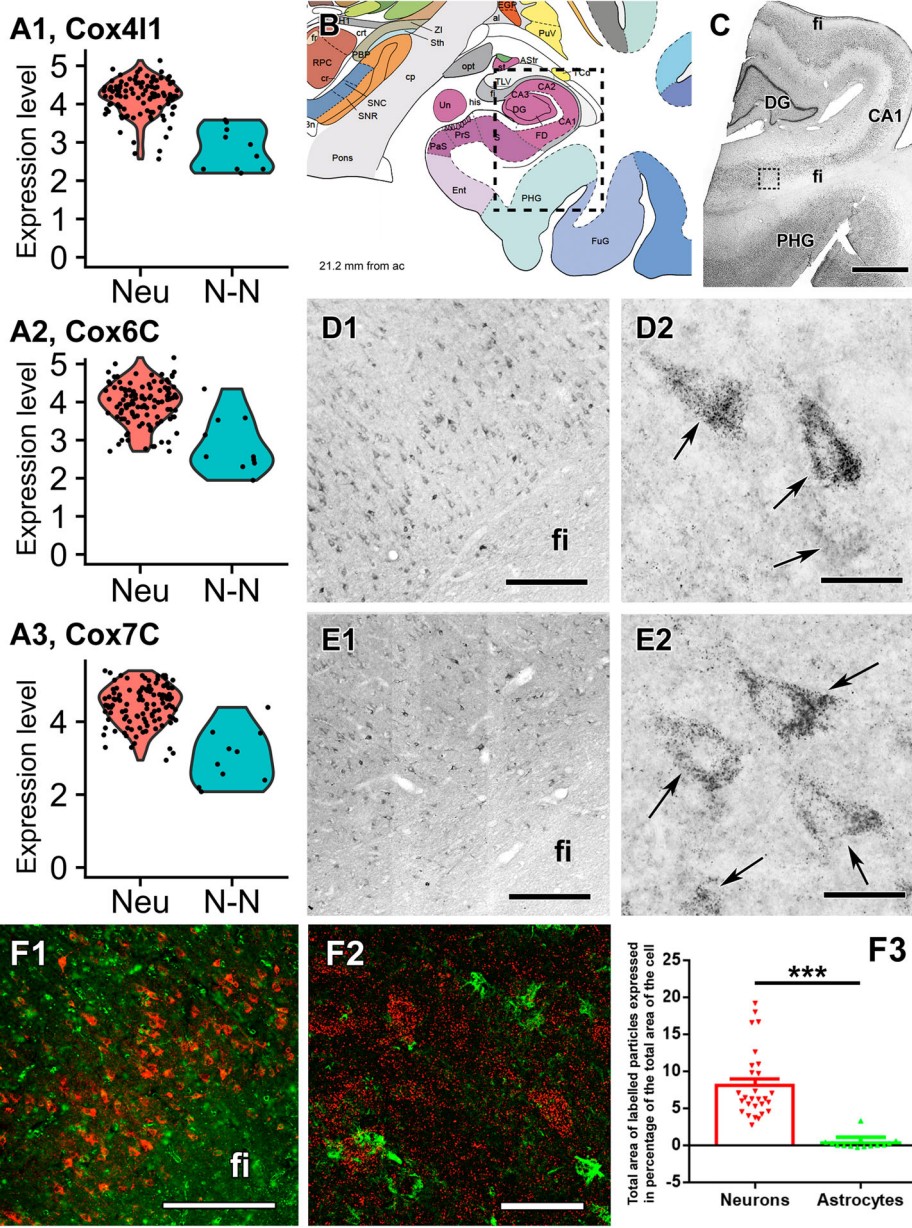

**Fig. 1 | The expression of CoxIV and cytochrome c in neuronal cells of the subiculum.** **A** Violin plots of expression of mRNA level in neuronal (Neu) and non-neuronal (N-N) cells based on single cell sequencing of Allen Institute for Brain Science. The dots represent individual cells. A1-3 panel demonstrate a dominantly neuronal expression of different CoxIV subunits. **B** Drawing of a coronal plane from a human brain atlas at 21.2 mm posterior to the anterior commissure (ac)[68]. **C** A Nissl stained section corresponding to the area framed in (**A**). **D** Parts of a CoxIV-immunolabelled coronal section, which is parallel to the section shown in (**B**). D1 shows the subicular area framed in (**C**) while D2 shows CoxIV-immunolabelled neurons at high magnification. **E** Parts of a Cytochrome c immunolabelled coronal section, which is parallel to the section shown in (**C**). E1 shows the area framed in (**C**) while E2 shows Cytochrome c immunolabelled cells at high magnification. In D2 and E2, arrows point to individual labelled cells, which are neurons based on their size and morphology. Also note the absence of labelling of cells with glial morphology and the dot-like distribution of labelling suggesting the location of the enzymes in mitochondria. In addition, note the absence of labelled cells in the white matter of fimbria hippocampi (fi). **F** Verification of non-astrocytic location of CoxIV (red) using double immunolabelling with the astrocyte marker S100B (green). F1: A low magnification image shows that only S100B but not CoxIV is present in the fimbria hippocampi (fi). F2: A higher magnification image demonstrates the lack of co-localization between CoxIV and S100. The high magnification image also shows the dot-like mitochondrial distribution of CoxIV. F3: Measuring the total area of CoxIV-labelled articles in percentage of the total area of the cells demonstrates intense CoxIV labelling in neuronal cell bodies while essentially no labelling in astrocytes, which leads to a profound statistical difference between the two cell types (***$p < 0.001$). Further abbreviations: CA1 = CA1 region of the hippocampus, DG Dentate gyrus, PHG Parahippocampal gyrus. Scale bars = 3 mm for (**C**), 300 μm for D1-F1, 50 μm for F2, 30 μm for D2 and E2, and 20 μm for F3. Source data are provided in Supplementary Data 1.

types in the subiculum (Fig. 5D−F), the dentate gyrus (Supplementary Fig. 1) and the parahippocampal cortex, too (Supplementary Fig. 2).

The MDH2 of the tricarboxylic acid cycle was also absent in human astrocytes but present in neurons in the subiculum (Fig. 6A), the dentate gyrus (Supplementary Fig. 1J) and the parahippocampal cortex (Supplementary Fig. 2D). IDH3 subunit A (a critical subunit of IDH3, the NADH-forming, mitochondrially localized isocitrate dehydrogenase) also exhibited a predominantly neuronal (on the basis of morphology) immunostaining (Supplementary Fig. 2I). Considering that human cortical astrocytes also lack ketoglutarate dehydrogenase complex-specific subunits[29], another major NADH-producing dehydrogenase, the need for harbouring OXPHOS is questioned. However, the group of Bolaños recently demonstrated that adult mouse astrocytes exhibit fatty acid oxidation (FAO) that organizes mitochondrial supercomplexes for sustaining reactive oxygen species formation and cognition[43]. The concept of astrocytic mitochondria catabolizing fatty acids has also been demonstrated in murine brain or cultures by the groups of Misgeld[44], Liu[45], Johnson[46] and Yin[47]. Readouts from other substrates have been examined by other groups specifically for astrocytes, but only from murine origin[48]. Only in the studies by the groups

of Malm[49], Erlandsson[50] and Langford[51] were the astrocytes from human origin, but they were either induced pluripotent cell (hiPSC)-derived astrocytes generated from neuroepithelial-like stem cells[50], or induced pluripotent stem cells of AD patients from skin biopsies or blood samples[49] or foetal brain tissues that were subject to culturing conditions for several days[51]. FAO generates reducing equivalents in abundance that could be oxidized by the respiratory chain. Here, we have not investigated whether FAO enzymes can be immunocytochemically detected in astrocytes of human brains. The group of Masgrau[52] inferred FAO in human astrocytic mitochondria by performing unbiased statistical comparisons of the mitochondrial transcriptomes in human (and mouse) astrocytes vs neurons using a database compiled by the group of Barres[53]. However, in ref. 52 the neuronal databases were available from only one subject. Thus, to the best of our knowledge, it is currently not known whether astrocytes in the adult human brain engage in FAO, and if yes, what is the fate of the reducing equivalents output. In turn, we found that SDH1B was present in both neuronal and astrocytic cell types in the subiculum (Fig. 6C), the dentate gyrus (Supplementary Fig. 1) and also in the parahippocampal cortex (Supplementary Fig. 2), which is expected, given the multiple roles of this

**Fig. 2 | Cytochrome C and CoxIV are located in neurons but not in astrocytes in the dorsomedial prefrontal cortex (DMPFC). A** Cytochrome C immunolabelling of a section of the DMPCF demonstrates immunolabelling of cells with neuronal morphology distributed in all layers. **B** High magnification of a cell pointed to by a black arrow in panel A shows dot-like mitochondrial immunolabelling of Cytochrome C in the cell body as well as the dendrites (see black arrowheads) leaving the cell body. **C** CoxIV-immunolabelled neurons are present in different layers of the DMPFC. **D** A high magnification image of a neuron indicated by the black arrow in (**C**). The dot-like mitochondrial immunolabelling is present in the cell body, the visible apical dendrite (black arrowhead) and the axon initial segment (white arrowhead) leaving the base of the pyramidal neuron. **E** Confocal image of a DMPFC section double labelled with CoxIV (green) and the astrocyte marker S100B (red). While neurons are labelled with CoxIV, astrocytes essentially do not contain CoxIV immunolabelling. **F** Quantification of the CoxIV labelling. The area occupied by immunolabelled particles within the cell body is very high in neurons but not in astrocytes, a markedly significant difference (***$p < 0.001$). Scale bars: 100 μm for A and C, 50 μm for E, and 20 μm for (**B, D**). Source data are provided in Supplementary Data 1.

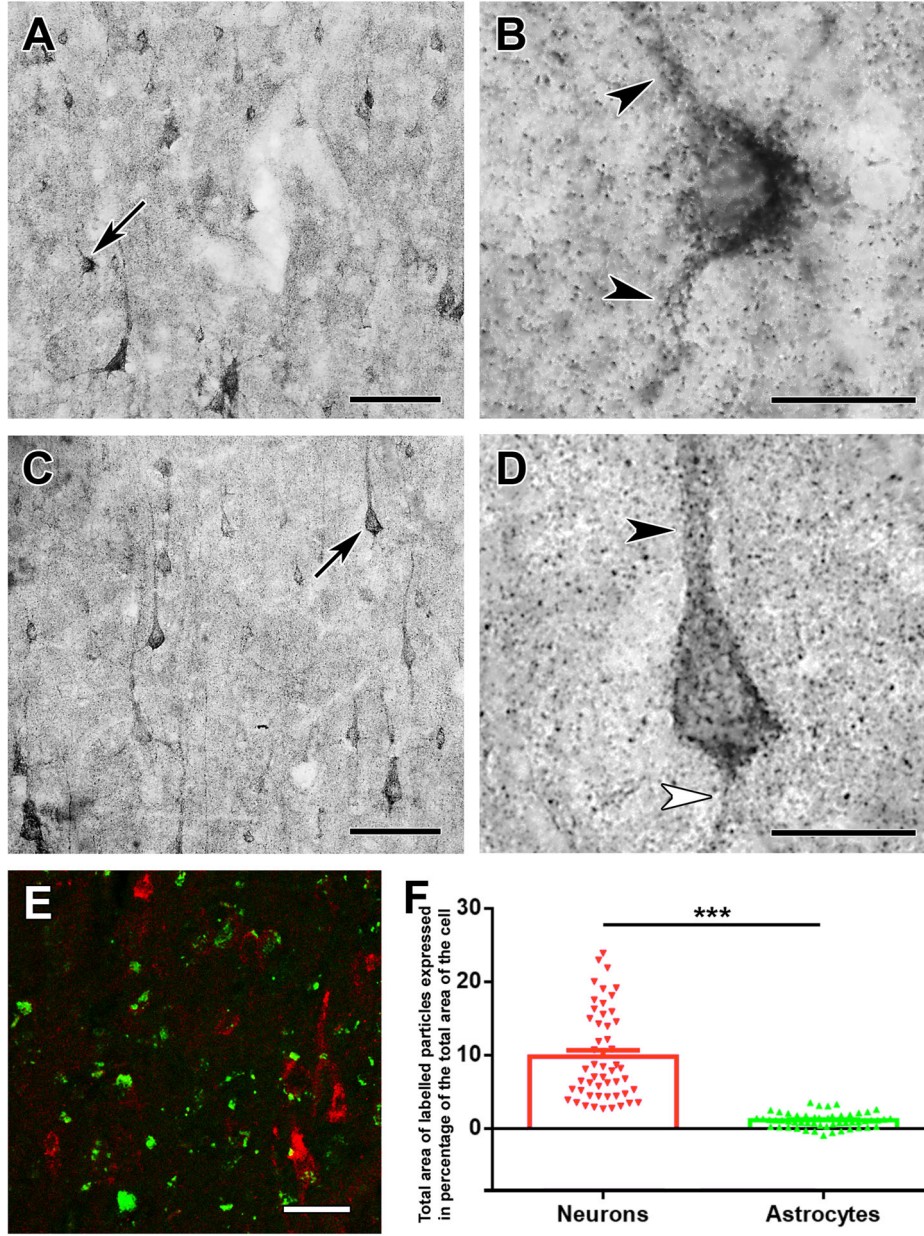

enzyme in non-OXPHOS pathways[54,55]. Nevertheless, the above evidence is compatible with the notion that $O_2$ consumption in the human cerebral cortex might be mostly attributed to neuronal mitochondria; mindful that in the human brain the glia-to-neuron ratio is close to one[56], it can be stated that close to- but less than half of the human brain is responsible for the 20% of the total body oxygen consumption[57], while only 1% of body weight. With the current state-of-the-art technological advances regarding in vivo functional assessment, it is not possible to verify that only neurons contribute significantly to $O_2$ consumption in the living human cerebral cortex. Even in the most recent, most advanced MRI of the human brain exhibiting a spatial resolution of 200 μm, 253–264 neuronal and glial bodies are included[58]. Having said that, it is notable that non-oxidative consumption of glucose during focal physiologic neural activity was reported in 1988[59], i.e. six years before ANLS hypothesis was formulated[1], that has led to the establishment of the concept of regional aerobic glycolysis in the human brain[60]. The lack of Cox IV from adult human cortical astrocytes should not be too surprising to the scientific community using rodents as laboratory animals: mice engineered to lack Cox IV in astrocytic mitochondria in vivo were fully viable in the absence of any signs of glial or neuronal loss even at 1 year of age[61]. Finally, it is important to consider the functional similarity between astrocytes and endothelial cells. Endothelial cells have evolved to perform minimal oxidative phosphorylation (OXPHOS) to ensure that $O_2$ in the blood can reach the most distal parts of the vasculature[62]. This mechanism is regulated by endogenous nitric oxide (NO) production[63]. Similarly, astrocytes, which contribute to the blood-brain barrier (BBB) around the brain microvasculature, should also avoid depleting $O_2$ from the interstitium (the neuronal parenchyma). Therefore, it is not surprising that human astrocytes exhibit a much lower capacity -if any- for OXPHOS, an $O_2$-consuming process. Moreover, the significantly higher presence of OXPHOS in murine astrocytes compared to human astrocytes in relation to the neurons of the same species can be understood by considering their relationship with the BBB. In humans, astrocytes form a much tighter barrier around the microvasculature by exhibiting a greater number of astrocytic end feet, than in murine tissues[64]. Given that the BBB is more permeable in murine brains, it is less critical for murine astrocytes to restrict OXPHOS to prevent $O_2$ depletion.

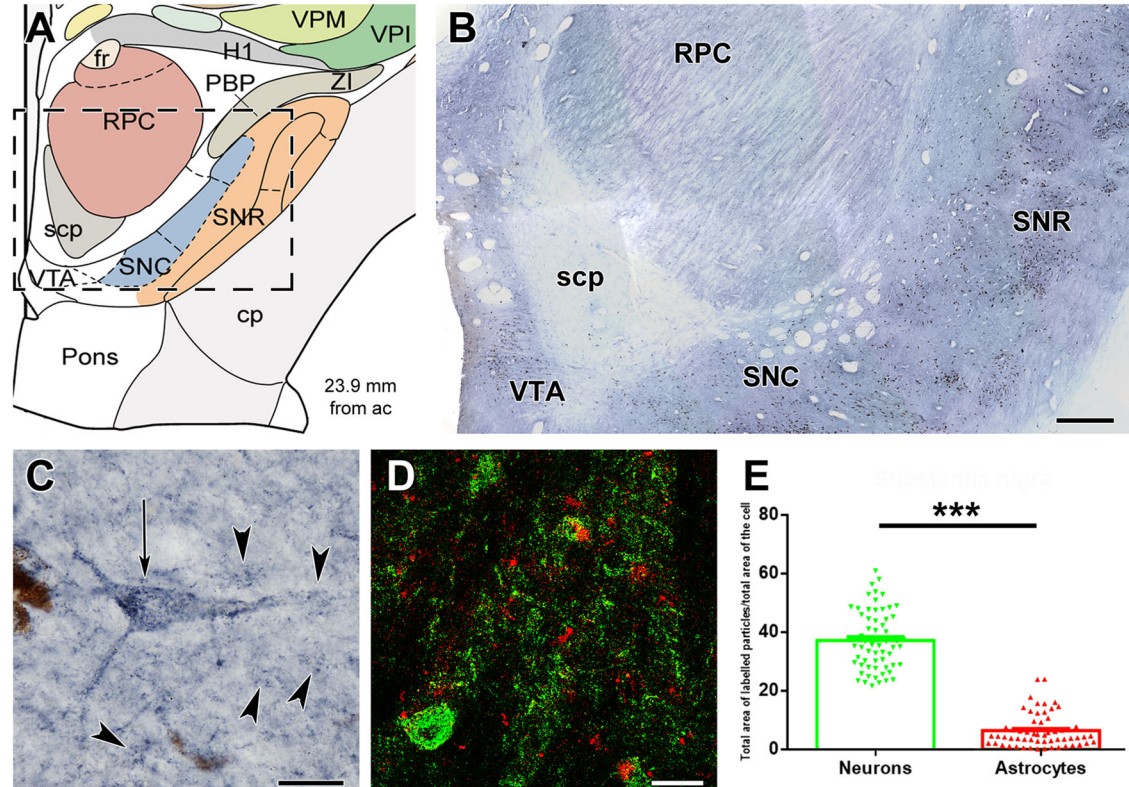

**Fig. 3 | CoxIV in the substantia nigra (SN). A** Drawing of a coronal plane at 23.9 mm posterior to the anterior commissure (ac)[68]. **B** A field corresponding to the framed area in A is shown immunolabelled for CoxIV. The dark cells in the ventral tegmental area (VTA) and substantia nigra (SN) are, however, neurons containing the dark matter characteristic of the regions. **C** High magnification of a cell pointed to by a black arrow shows dot-like mitochondrial immunolabelling of CoxIV in the cell body as well as the dendrites. The arrowheads point to presumed astrocytes expressing a low level of CoxIV. D: CoxIV-immunolabelled neurons are present in different layers of the DMPFC. **D** Confocal image of a SN section double labelled with CoxIV (green) and the astrocyte marker S100B (red). While neurons are labelled intensively with CoxIV, astrocytes also contain some CoxIV immunolabelling. **E** Quantification of the CoxIV using the S100B double labelling. The area occupied by immunolabelled particles within the cell body is very high in neurons. Astrocytes also contain CoxIV immunoreactivity in a significantly lower amount (***$p < 0.001$). RPC Parvocellular subdivision of the red nucleus, scp Superior cerebellar peduncle, SNC SN, pars compacta, SNR SN, pars reticulata. Scale bars: 1 mm for (**B**), 30 µm for (**C**), and 50 µm for (**D**). Source data are provided in Supplementary Data 1.

## Limitations of the study

Here we show that astrocytes within formations of adult human brains exhibit barely detectable levels of enzymes that are critical for OXPHOS. The detection methods were amplified immunohistochemistry and imaging mass cytometry. Additionally, we compiled cell-specific transcriptomic data from the Allen Institute for Brain Science. Our findings suggest that astrocytes in the adult human brain are likely unable to perform OXPHOS, which could support the Astrocyte-To-Neuron Lactate Shuttle (ANLS) model. However, we do not provide functional evidence of the absence of oxidative metabolism in these cells. It is important to note that demonstrating cell-specific oxidative metabolism within the human brain is currently unfeasible. While cell-specific OXPHOS can be shown in cell cultures or live murine brains, as extensively documented by others and referenced herein, such evidence for astrocytes within the human brain remains elusive.

## Methods
### Human brain tissue samples used for immunohistochemistry
Human brain samples were collected in the Human Brain Tissue Bank, Semmelweis University in accordance with the Ethical Rules for Using Human Tissues for Medical Research in Hungary (HM 34/1999) and the Code of Ethics of the World Medical Association (Declaration of Helsinki). Tissue samples were taken during a brain autopsy at the Department of Forensic Medicine of Semmelweis University in the framework of the Human Brain Tissue Bank (HBTB), Budapest. The activity of the HBTB has been authorized by the Committee of Science and Research Ethic of the

Ministry of Health Hungary (ETT TUKEB: 189/KO/02.6008/2002/ETT) and the Semmelweis University Regional Committee of Science and Research Ethic (No. 32/1992/TUKEB). The study reported in the manuscript was performed according to a protocol approved by the Committee of Science and Research Ethics, Semmelweis University (TUKEB 189/2015). The medical history of the subjects was obtained from clinical records, interviews with family members and relatives, as well as pathological and neuropathological reports. All personal data are stored in strict ethical control, and samples were coded before the analyses of tissue.

Human brain tissue samples used for imaging mass cytometry were sourced from the Newcastle Brain Tissue Resource, UK, with written consent given by the donors or their next to kin for research purposes. Ethical approval for the use of all tissues was provided by the National Health Service Local Research Ethics Committee and adhered to the Medical Research Council, UK's guidelines regarding the use of human tissue in medical research.

### Tissue collection for immunolabelling
Immunohistochemistry was used to assess the location of mitochondrial proteins and identify cells as neurons or astrocytes using a neuronal vs astrocytic marker, respectively. For immunolabelling hippocampus and the adjacent portion of the parahippocampal gyrus, brain blocks from a 62 years old female and a 56 years old male individual were used. In turn, immunolabelling of the dorsomedial prefrontal cortex (DMPFC) and the substantia nigra (SN) was performed using samples of a 58 year old male, while

**Fig. 4 | The expression of CoxIV in the CA1 region of the hippocampus measured by imaging mass cytometry. A** Example of an image demonstrating CoxIV (blue), HuB+C + D (red) as neuronal marker, and glial fibrillary acidic protein (green) as a marker of astrocytes. **B** The red channel of the image in panel A shows neuronal labelling. The borders of three selected neurons are delineated by yellow. **C** The green channel of the image in (**A**) shows labelling of astrocytes. The borders of three selected astrocytes are delineated by yellow. **D** Quantification of CoxIV labelling with identified neurons and astrocytes. The amount of CoxIV labelling per area unit is significantly (***$p < 0.001$) higher is neurons than is astrocytes. **E** CoxIV labelling (blue channel in **A**) with the 3 neurons delineated in (**B**) indicated by yellow lines. **F** CoxIV labelling (blue channel in **A**) with the 3 astrocytes delineated in (**C**) indicated by yellow lines. Scale bar = 300 μm. Source data are provided in Supplementary Data 1.

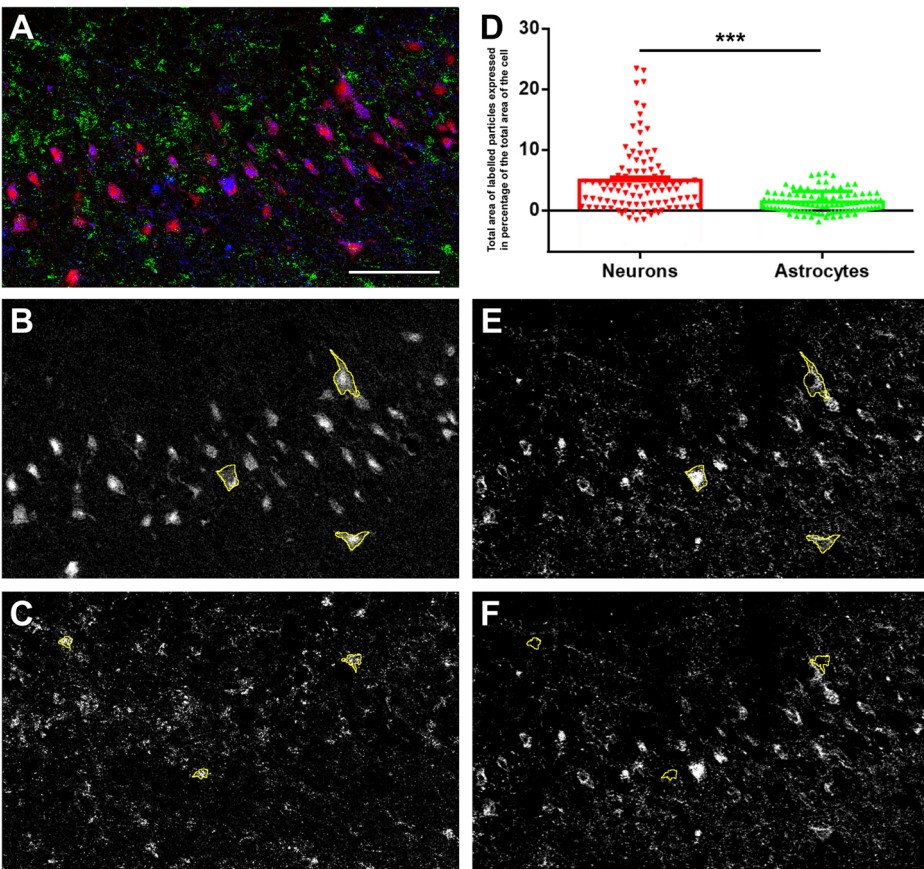

brain sections of a 60 years old female were used for immunolabelling of the anterior cingulate cortex. The donors had no brain-related disorders. Post-mortem delay prior to obtaining the samples was 12, 4, 22 and 48 hours, respectively. Further information regarding these human brains can be obtained from the Human Brain Tissue Bank of the Semmelweis University (https://semmelweis.hu/hbtb/). The blocks were cut into 10 mm thick coronal slices and immersion fixed in 4% paraformaldehyde in 0.1 M phosphate-buffered saline (PBS) for 5 days. Subsequently, the block was transferred to PBS containing 0.1% sodium azide for 2 days to remove excess paraformaldehyde. Then, the block was placed in PBS containing 20% sucrose for 2 days of cryoprotection. The block was frozen and cut into 60 μm thick serial coronal sections on a sliding microtome. Sections were collected in PBS containing 0.1% sodium azide and stored at 4 °C until further processing.

### DAB immunolabelling

Free-floating brain sections were immunolabeled for mitochondrial proteins (see antibodies and their dilutions in Suppl. table 1). The antibodies were applied for 24 h at room temperature, followed by incubation of the sections in biotinylated anti-rabbit/mouse secondary antibody (1:1,000 dilution, Vector Laboratories, Burlingame, CA) and then in avidin-biotin-peroxidase complex (1:500, Vector Laboratories) for 2 h. Subsequently, the labelling was visualized by incubation in 0.02% 3,3-diaminobenzidine (DAB; Sigma), 0.08% nickel (II) sulphate and 0.001% hydrogen peroxide in PBS, pH 7.4 for 5 min. Sections were mounted, dehydrated and coverslipped with Cytoseal 60 (Stephens Scientific, Riverdale, NJ, USA).

### Double labelling with the astrocytic marker S100B and the neuronal marker HuC/HuD (HuB+HuC+HuD)

Double immunofluorescence staining was used to clarify the colocalization of mitochondrial genes with the astrocytic marker S100B and the neuronal

marker HuC/D in the hippocampus and parahippocampal gyrus. To reduce autofluorescence, tissue sections were treated with 0.15% Sudan Black B (in 70% ethanol) after antigen retrieval (10 min at 90 °C in 0.05 M Tris buffer, pH = 9.0) procedures. Slides were blocked by incubation in 3% bovine serum albumin (with 0.5% Triton X-100 dissolved in 0.1 M PB, Sigma) for 1 h at room temperature, followed by washing with washing buffer (10 min × 3). Mitochondrial proteins were immunolabeled using the same antibodies as for single labelling. The visualization was performed with incubation in Alexa 594 donkey anti-rabbit IgG secondary antibody (1:500 dilution, Vector Laboratories) for 1 h. Subsequently, sections were placed in mouse anti-S100B (1:250 dilution, Sigma, Cat. No. S2532-1) for 24 h at room temperature. The sections were then incubated in Alexa 488 donkey anti-mouse IgG secondary antibody (1:500 dilution, Vector Laboratories). Alternatively, Alexa 488 donkey anti-rabbit and Alexa 594 donkey anti-mouse IgG secondary antibodies were used, respectively. Finally, all sections with fluorescent labels were mounted on positively charged slides (Super-frost Plus, Fisher Scientific, Pittsburgh, PA) and coverslipped in antifade medium (Prolong Antifade Kit, Molecular Probes). The RRIDs and other identifiers of the mitochondrial protein-specific antibodies, anti-S100B and anti- HuC/D (HuB+HuC+HuD) antibodies are summarized in Suppl. Table 1.

### Microscopy and photography

Sections were examined using an Olympus BX60 light microscope also equipped with fluorescent epi-illumination. Images were captured at 2048 × 2048 pixel resolution with a SPOT Xplorer digital CCD camera (Diagnostic Instruments, Sterling Heights, MI, USA) using a 4× objective for dark-field images, and 4–40× objectives for bright-field and fluorescent images. Confocal images were acquired with a Zeiss LSM 70 Confocal Microscope System using a 40-63X objectives at an optical thickness of 1 μm for counting varicosities and 3 μm for counting labelled cell bodies. The

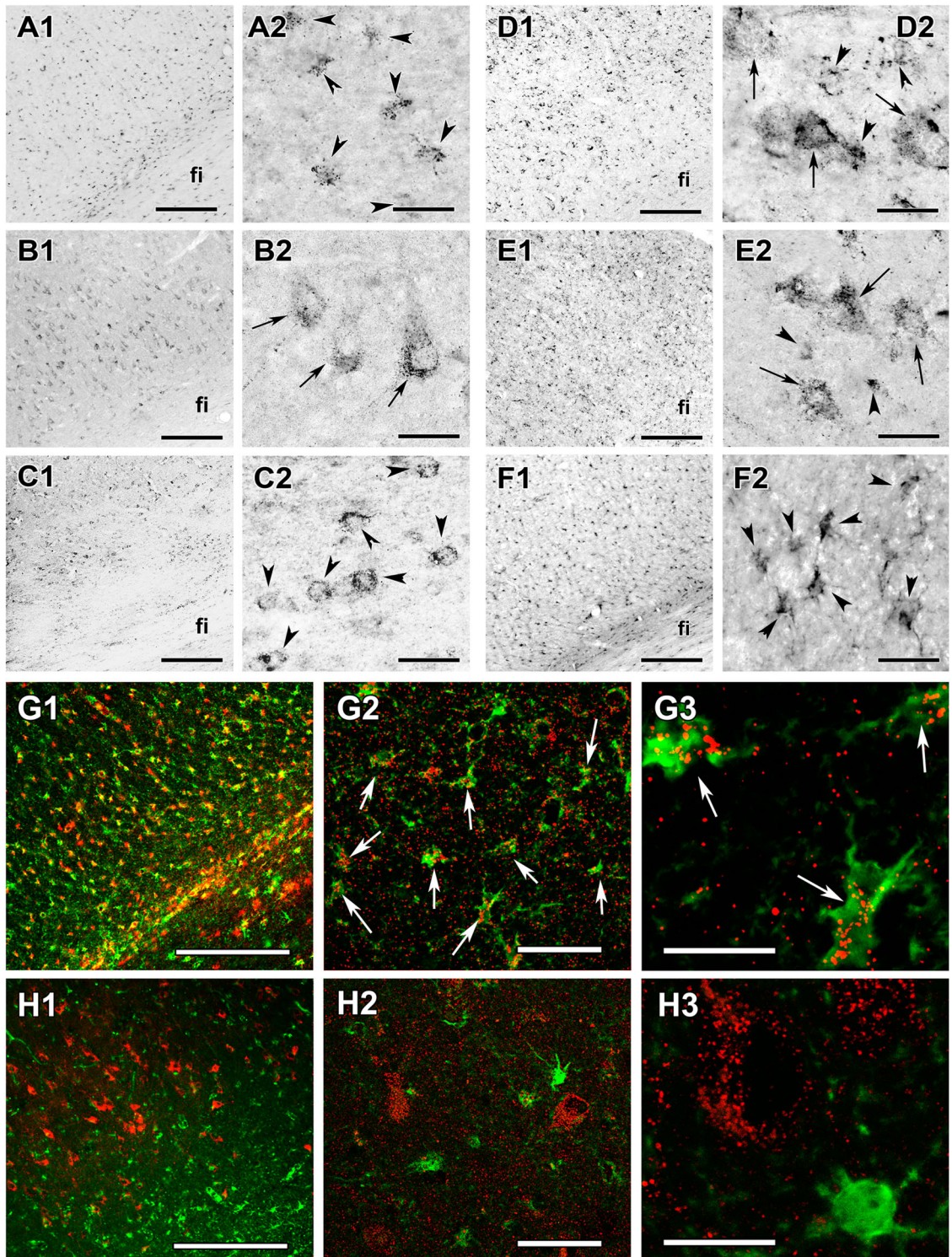

contrast and sharpness of the images were adjusted using the 'levels' and 'sharpness' commands in Adobe Photoshop CS 8.0. Full resolution was maintained until the photomicrographs were cropped, at which point the images were adjusted to a resolution of at least 300 dpi.

**Quantification of double immunolabelled images**

The borders of neuronal cell bodies were identified based on immunolabelling using HuC/D (see below) or with the mitochondrial OXPHOS markers (as indicated in the figures) while the borders of astrocytes were determined using S100B immunolabelling. Then, the area of the cell bodies on the sections was determined using ImageJ (Rasband, W.S., ImageJ, U. S. National Institutes of Health, Bethesda, Maryland, USA, https://imagej.net, 1997–2018). The area occupied by the mitochondrial labelling was also determined with ImageJ, and the density of the labelling was determined as the amount of mitochondrial labelling expressed in percentage of the area of the cell body (Supplementary Fig. 5). To eliminate the possibility that the identification of the neurons based on the mitochondrial OXPHOS marker creates bias, mitochondrial expression levels were also measured in neurons identified by the neuronal marker HuC/D protein. These measurements resulted in a very

**Fig. 5 | Immunolabelling of pyruvate carboxylase, MPC1, MPC2, and PDH enzymes in the subiculum.** The same brain areas are shown in Fig. 1. The left panels indicate pictures taken at low while the right panels next to them taken at high magnification. In the high magnification images, arrows point to neurons while arrowheads point to astrocytes. **A** Parts of a pyruvate carboxylase immunolabelled coronal section. Arrowheads in A2 point to individual labelled cells, which resemble astrocytes based on the relatively small size of their cell bodies as well as their morphology. Note the dot-like distribution of labelling consistent with the mitochondrial location of the enzymes. **B** MPC1-immunolabelled sections show labelled neurons based on their exclusive distribution in the grey matter (see the absence of labelled cells in the white matter of fimbria hippocampi (fi) in B1 as opposed to their presence in A1), and also on their size and morphology (B2). **C** MPC2-immunolabelled section. Labelled cells are present in the white matter of the fi (C1). The high magnification picture (C2) shows a relatively small size of labelled cells, likely astrocytes. **D** PDHE1b, **E** PDHE2, **F** PDHX. Low magnification pictures on the left (D1, E1, F1) as well as high magnification figures on the right show that PDHE1b and PDHE2 are expressed in both neurons and astrocytic cells based on the

morphology of labelled cells while PDHX expression is exclusively astrocytic. **G** Demonstration of the astrocytic location of pyruvate carboxylase (red) using double immunolabelling with the astrocyte marker S100B (green). Relatively low magnification image (G1) shows that S100B and pyruvate carboxylase are present in the grey as well as in the white matter (fimbria hippocampi - fi). The higher magnification image (G2) demonstrates that all cells are double labelled with pyruvate carboxylase and S100B (pointed to by white arrows). The high magnification image in the right panel (G3) shows the dot-like mitochondrial distributions of pyruvate carboxylase. Astrocytes visualized by S100B labelling (green) contain pyruvate carboxylase-containing mitochondria (red). **H** Demonstration of the non-astrocytic location of MPC1 (red) using double immunolabelling with the astrocyte marker S100B (green). The low magnification image in the left panel (H1) shows that S100-immunolabelled are present in the white matter of fi while MPC1-immunoreactivity is absent in this region. The higher magnification images demonstrate the lack of double labelling between MPC1 and S100B (H2, H3). Scale bars: 300 μm for A1−H1, 50 μm for G2 and H2, 30 μm for A2−F2, and 20 μm for G3 and H3.

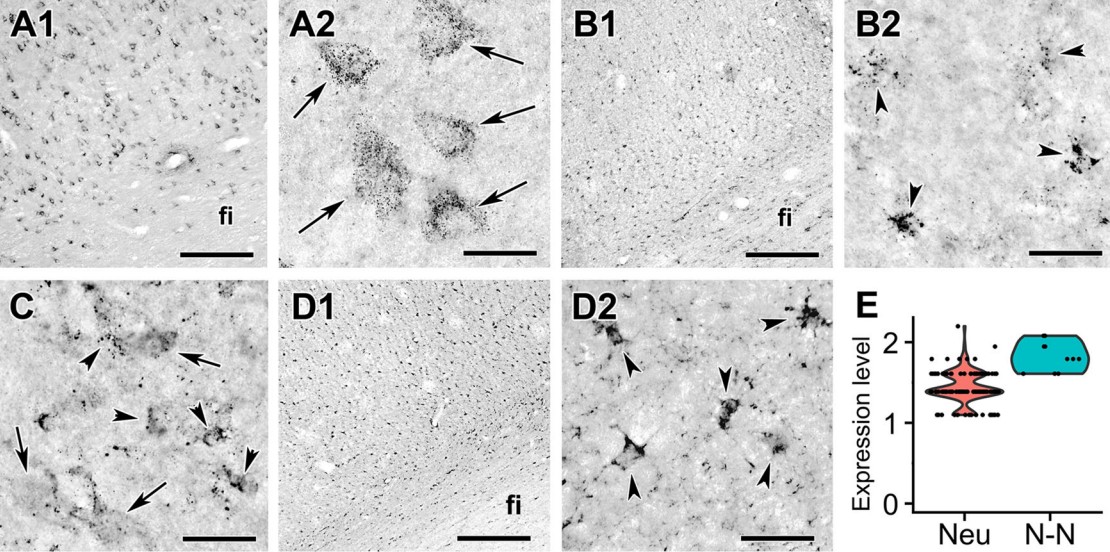

**Fig. 6 | Labelling of mitochondrial dehydrogenases in the subiculum.** Arrows point to neurons while arrowheads indicate astrocytic cells. Note the dot-like mitochondrial labelling. **A** Low magnification picture on the left (A1) as well as the high magnification figure on the right (A2) show that MDH2 is expressed exclusively in neurons. **B** Low magnification image on the left (B2) as well as high magnification image on the right show that IDH2 expression is exclusively astrocytic. **C** A high magnification image demonstrated that SDH1-B is expressed in both neurons and

astrocytic cells based on the morphology of labelled cells. **D** SDH-C expression is exclusively astrocytic. **E** Violin plot of expression of mRNA level in neuronal (Neu) and non-neuronal (N-N) cells based on single cell sequencing from Allen Institute for Brain Science. The dots represent individual cells demonstrating a dominantly non-neuronal expression of SDH-C. Scale bars: 300 μm for A1, B1 and D1, and 30 μm for A2, B2, C and D2. Source data are provided in Supplementary Data 1.

similar level of neuronal labelling as when neurons were identified by the mitochondrial OXPHOS markers (Supplementary Fig. 6).

## Imaging mass cytometry

Formalin-fixed paraffin-embedded sections (5 μm) of hippocampus at CA1 region obtained from 8 donors that exhibited no brain-related disorders (Post-mortem delay: range from 16-66 h, mean ± SD = 42.48 ± 16.62) were subjected to imaging mass cytometry, a single-cell multiplex imaging technique using Lanthanide-conjugated antibodies. Further information regarding the donors and the brain samples can be obtained from the Newcastle Brain Tissue Resource at https://nbtr.ncl.ac.uk/. The establishment of the antibody panel and labelling of sections for imaging mass cytometry detection were performed as described in ref. 31. In this study, anti-MTCO1 (Clone 1D6E1A8; conjugated to 158Gd; dilution 1:50; ab14705, Abcam) was used to label OXPHOS complex IV. Cell markers, including anti-HuB+C + D antibody (Clone 16A11C1D4; conjugated to 155Nd; dilution 1:500; ab176106, Abcam) for neurons; anti-GFAP (Polyclonal, conjugate to 141Pr; dilution 1:500; Agilent Z0334) for astrocytes, and

anti-Histone H3 (Clone 24HC2LC12; conjugated to 142Nd; dilution 1:400; 701517, ThermoFisher) for nuclei were also employed.

## Analysis of transcripts deposited in the Allen Institute for Brain Science coding for mitochondrially-localized proteins in individual brain cells

The gene symbols of mitochondrially-localized subunits of proteins enlisted in MitoCarta 2.0[65] curated to include newer data reported in MitoCarta 3.0[66] were filtered to include only those that are involved in OXPHOS (respiratory chain components, transporters of adenine nucleotides, Pi and substrates, and the citric acid cycle dehydrogenases MDH2, IDH2 and IDH3). Subsequently, they were queried using the Transcriptomics Explorer of the Allen Institute for Brain Science human and mouse brain transcriptome (RNA-seq) repository (https://portal.brain-map.org/atlases-and-data/rnaseq). Regarding human tissues, brain samples (~500 samples from 107 brains covering the middle temporal gyrus (MTG), anterior cingulate gyrus (CgGr), primary visual cortex (V1C), primary motor cortex (M1C), primary somatosensory cortex (S1C) and primary auditory cortex (A1C)) from

either postmortem or neurosurgical origin were made available through tissue donors. The total number of individual human brain cells sorted is 49,495. Clinical summaries and donor characteristics as well as a description of the criteria for acceptance of use are provided in https://tinyurl.com/Allenhumanprotocols. Regarding mouse tissues, samples were obtained from 86 adult (postnatal day P53-P59) mice, both male and female, from various regions of the mouse brain. The total number of individual mouse brain cells sorted is approximately 4 million. Further details regarding tissue preparation, single nucleus dissociation, sorting, RNA-sequencing and cell type clustering are provided in https://portal.brain-map.org/atlases-and-data/rnaseq/protocols-human-cortex and https://portal.brain-map.org/atlases-and-data/rnaseq/protocols-mouse-cortex-and-hippocampus for human and mouse tissues, respectively. Ultimately, data were visualized as heatmaps of mRNA expression levels in neuronal (Neu) and non-neuronal (N-N) cells. Heatmaps (homo for human, expressed as measure of central tendency, trimmed means, 25−75%, log2 (CPM + 1), range 0−16.61, and mus for mouse, expressed as measure of central tendency, median, log2 (CPM + 1), range 0−17.89) are shown per mitochondrial OXPHOS entity or functional unit in Supplementary Data 2. Names of genes coding for individual subunits and/or isoforms and/or pseudogenes are shown on the left y-axis. See for example the heatmap in Supplementary Fig. 7: names of genes coding for complex IV subunits in humans are indicated on the left y-axis. On the top part of the x-axis, individual NeuN-positive nuclei (originating from neurons) versus NeuN-negative (originating from non-neuronal cells including astrocytes), segregated according to the clustering algorithm detailed in https://portal.brain-map.org/atlases-and-data/rnaseq/protocols-human-cortex are depicted. Cell clustering is outlined by the lines branching on the top of the x-axis. Abbreviations are given in https://portal.brain-map.org/atlases-and-data/rnaseq. Non-neuronal elements are shown in the far right of the heatmap, indicated as 'Non-neuronal' at the bottom (orange). It is visually striking that transcripts coding for complex IV subunits in human brain cells (and practically for all OXPHOS components, see Supplementary Data 2) are almost undetectable in non-neuronal elements compared to neuronal elements as indicated at the bottom of the heatmap (green). This stark contrast between neuronal and non-neuronal elements is not observed in transcripts coding for complex IV subunits for mouse brain cells, shown in Supplementary Fig. 8, and the pattern is similar for practically all OXPHOS components, see Supplementary Data 2). The Supplementary Data 2 contain the analysis of transcripts coding for all mitochondrially-localized proteins critical for OXPHOS in humans and mouse brains, respectively. Heatmap range is identical for all panels as shown in the last panels in either dataset depicting Pi carriers and dicarboxylate carriers.

## Statistics and Reproducibility

Data are presented as averages ± SEM. Significant differences between two groups were evaluated by Student's t-test or Mann–Whitney U Test if normality failed. The total number of brains used were from 4 donors for immunohistochemistry, 8 donors for imaging mass cytometry and 107 brains for transcriptomic analysis. The number of the cells evaluated per view field is indicated in the respective legends or main text.

## Data availability

All data have been deposited at Mendeley (https://doi.org/10.17632/wj3ndgbchf.1)[67]. Source data are provided in Supplementary Data 1. All other data are also available through the corresponding author (Christos Chinopoulos) on reasonable request.

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

## Acknowledgements
This work was supported by grants from NKFIH KH129567, NKFIH K135027 and TKP2021-EGA-25 to Christos Chinopoulos, the Michael J. Fox foundation (grant 15707), the Parkinson's UK (grant G2003) and the Wellcome Centre for Mitochondrial Research (203105/Z16/Z) to Chun Chen, NKP-2017-00002 to M.P., and NKFIH OTKA K134221 and K146077 and HAS-NAP2022-I-3/2022 to A.D. Brain tissues provided by the Newcastle Brain Tissue Resource (NBTR) were funded by the UK Medical Research Council (G0400074), NIHR Newcastle Biomedical Research Centre and Unit and the Alzheimer's Society and Alzheimer's Research Trust. The generation of mass cytometry data was supported by the Newcastle University Flow Cytometry Core Facility.

## Author contributions
A.D., M.C., and Chun Chen performed the experiments. A.B., Chun Chen, M.P. and G.H. provided samples. A.S., K.P. and J.L. performed data analysis. A.D. edited the manuscript. Christos Chinopoulos outlined the experiments, performed data analysis and wrote the manuscript.

## Funding

## Competing interests
The authors declare no competing interests.
