## [Peer Review File · Communications Biology]

Reviewers' comments:

Reviewer #1 (Remarks to the Author):

The manuscript by Dobolyi and colleagues analysis oxidative metabolism in human astrocytes, which is of very high interest and significance. The authors claims that adult human neocortex and hippocampal astrocytes hardly express proteins related to oxidative metabolism.

However, the study is based only in immunochemical measurements with few individuals and with methodological analysis that raise some concerns. Moreover, the authors ignore a considerable amount of data in the literature demonstrating oxidative metabolism in astrocytes in human and murine cell lines and tissues. Thus, there are weaknesses that dampen enthusiasm for this manuscript in the current format.

My concerns for this manuscript are as follows:

- The authors only mention the literature from research groups supporting their own hypothesis, without mention publications demonstrating the existence of oxidative metabolism coming from different substrates in human astrocytes (Eraso-Pichot et al., 2018; Kottinen et al., 2019; Natarajaseenivan et al., 2018; Potter et al., 2019; Zysk et al., 2023, as for example) and murine astrocytes (Azarias et al., 2011; Ioannu et al., 2019; Farmer et al., 2019; Qi et al., 2021 to give few examples) and using a wide range of assays, including enzymatic activity and oxygen consumption. Moreover, in the introduction, authors only cite old data from Bolaños group when their last paper demonstrate fatty acid mitochondrial metabolism (Morant-Ferrando et al., 2023). All these studies need to be taken into account and results presented in the present manuscript need to be compared to the previous literature.
- In general, methodology is not clearly described not justified. Some concerns on the labelling and quantification methodology are: could the different size of neurons vs astrocytes alter the quantifications as measurements are normalized by area? Importantly, according to methods (lines 230-232), the detection of neuronal and astrocytic borders were performed through mitochondrial markers for the first case and astrocytic markers for the second. The different criteria could clearly alter the measurements. Experiments should be performed in neuronal and astrocytic markers to clearly specify the borders of each cell type.
- Supplementary datasets of mRNA of OXPHOS should be clearer explained (subjects, exact methodology) and discussed, as they constitute key data.
- Lines 100-101: authors state that they use a large sample but not characteristics of the subjects are given.
- Lines 100-104: COXIV expression levels are around 4 times lower in astrocytes compared to neurons in CA1 regions and around 9 times in astrocytes. Authors conclude and stress in the abstract that “hippocampal astrocytes express evanescent amounts of mitochondrial proteins”. This overinterpretation of data is repeated through all the manuscript. First there is expression of this enzyme in astrocytes. Second, such differences might not be translated to any significant difference in oxidative metabolism between these two cell types.
- Quantifications should be provided in figure 5

- Why the expression of MDH2 is not analyzed in all the brain areas that COXIV was analyzed?
- Lines 140-145: authors state “it is safe to assume that O₂ consumption in the human cerebral cortex should be attributed almost entirely to neuronal mitochondria” based on an expression of MDH2 in some areas and no reported expression of KGDHC. To my point of view this is an overstatement because they do not consider all brain area neither fatty acid oxidation that produces NADH and FADH₂ per se.
- Lines 153-156: authors sound patronizing and to my point of view they are misinterpreting their own data. They state there is no COXIV in brain astrocytes, whereas their data show that there is: “The percentage of the cell body covered by COXIV-immunolabelled particles was 7.82 ± 0.88 (n=28 cells) and 9.83 ± 0.84 (n=52 cells) for neurons in the two brain areas, respectively. In contrast, these values were 0.32 ± 0.18 (n=18 cells) and 1.14 ± 0.14 (n=52 cells) for astrocytes”. “In the human substantia nigra, we found a considerably higher level of Cox IV immunoreactivity in astrocytes compared to the other brain regions, albeit still much less than in neurons (Fig. 3). The percentage of the cell body covered by COXIV-immunolabelled particles was 37.18 ± 1.24 (n=64 cells) for neurons in the substantia nigra while for astrocytes, the value was 6.34 ± 0.76 (n=59 cells).” All these values indicate protein expression and values of substantia nigra astrocytes are higher than values of CA1 neurons. Other authors have previously reported expression of COXIV in substantia nigra human astrocytes (Chen et al., 2021).
- The terms of glia and astrocytes are equally used, whereas they are not synonyms.
- I would like to stress that authors only provide data of expression by immunolabelling of some enzymes, but their conclusions point to the non-existence of oxidative metabolism. Despite the disagreements previously reported (there is expression of enzymes in astrocytes), without functional data, authors cannot disclose the existence of oxidative metabolism in human astrocytes. Caution in all their conclusions needs to be taken.

Reviewer #2 (Remarks to the Author):

In this work, Dobolyi and colleagues have performed an in-depth investigation of the occurrence and abundances of key mitochondrial proteins involved in the tricarboxylic acid (TCA) cycle and oxidative phosphorylation (OXPHOS) in several brain regions of post-mortem human brain. The authors performed this study with the particular interest to compare the levels of expression of these proteins, as potential indicators of their activities, in neuronal versus non-neuronal or astrocytic cells with the idea to provide evidence for or against the astrocyte-neuron lactate shuttle (ANLS) model. The sample size used is adequate for this type of study, the validation of the antibodies and the quantification approaches of the images are correctly performed. They found consistent data strongly suggesting a higher expression of these proteins in neurons versus astrocytes, leading them to suggest that in the human brain these observations are compatible with the ANLS model.

Comments

1. Lines 22-28 – Abstract. Whilst the main notion of the work is summarized in this section, this reviewer has some suggestions for amendments aimed to try to accurately obtain the conclusions specifically obtained from this work. The suggested abstract text is as follows:

“The astrocyte-to-neuron lactate shuttle model entails that, upon glutamatergic neurotransmission, glycolytically derived pyruvate in astrocytes is mainly converted to lactate instead of being entirely catabolized in mitochondria. The mechanism of this metabolic rewiring and its occurrence in human brain are unclear. Here by using immunohistochemistry and imaging mass cytometry we show that astrocytes of the adult human neocortex and hippocampal formation express evanescent amounts of mitochondrial proteins critical for performing oxidative phosphorylation (OXPHOS). With less OXPHOS, human brain astrocytes are thus bound to produce more lactate to avoid interruption of glycolysis.”

2. In general, throughout the manuscript the authors use strong assertions such as “lack of ...”. For instance, they used “lack of Cox IV”. However, it would be more accurate to read “undetected Cox IV” or a similar statement that does not imply negation. Any negating statement should be avoided given that they rely on the sensitivity and specificity of the techniques used, so there are not absolute assertions. Please, amend the manuscript accordingly.

3. Lines 131 – ref. 43 refers to MCT1 (monocarboxylate transporter-1), not to Mitochondrial Pyruvate Carrier-1 (MPC1) as the authors determined in this manuscript.

4. Lines 140-141 – “This, together with the fact that human cortical astrocytes do not express KGDHC(29) essentially mitigates any reasoning for having OXPHOS in them.” This sentence is an over-interpretation of the data. The absence of labeling of some TCA cycle enzymes does not directly demonstrate lack of TCA activity and, furthermore, does not imply no need for OXPHOS. There are alternative metabolic sources for NADH and FADH in astrocytes that could feed the OXPHOS such as, e.g., the fatty acids mitochondrial β -oxidation (Morant-Ferrando et al., Nat Metab. 5, 1290-1302, 2023; doi: 10.1038/s42255-023-00835-6.). The authors should mention this possibility.

5. Lines 142-142 – “Therefore, it is safe to assume that O₂ consumption in the human cerebral cortex should be attributed almost entirely to neuronal mitochondria”. Whilst the data provided in this manuscript and the previously published evidence might be compatible with this notion, this assertion is rather speculative and should be tuned down. For instance, the authors could consider stating something like “Therefore, this evidence is compatible with the notion that O₂ consumption in the human cerebral cortex might be mostly attributed to neuronal mitochondria.”.

6. Lines 157-159 – This paragraph seems to be misplaced as stands. The authors should consider placing this observation somewhere else within the R&D section, but not at the end of the section after the major conclusions were described.

Reviewer #3 (Remarks to the Author):

The paper of Dobolyi and co-workers reports that astrocytes of both the adult human neocortex and the hippocampal formation express low level of mitochondrial proteins critical for performing oxidative phosphorylation (cytochrome c and Cox IV) as compared to neurons. Conversely, astrocytes express high level of pyruvate carboxylase as compared to neurons. Authors used immunohistochemistry and imaging mass cytometry in a relatively low number of samples to conclude that human brain astrocytes are bound to produce lactate to avoid interruption of glycolysis, therefore supporting the astrocyte-to-neuron lactate shuttle hypothesis put forward in 1994 by Luc Pellerin and Pierre Magistretti based on work carried out on mouse astrocytes in culture.

Numerous teams using different models have tested this hypothesis experimentally. A consensus in the field has emerged that neurons, unlike astrocytes, are not equipped to effectively regulate glycolytic flux. The efficiency of the respiratory chain, on the other hand, is much better in neurons. This ability to regulate glycolytic flux by astrocytes enables them to carry out a number of anabolic reactions, and is not assessed solely in terms of energy.

The data obtained in this study do not significantly advance knowledge in this field, apart from the fact that it concerns human samples for which it is more difficult to carry out functional studies. Unfortunately, the study of enzyme expression alone (molecular or protein) does not allow us to advance this scientific question in any meaningful way.

I also have two methodological questions that temper my enthusiasm.

1. The number of samples seems to me too small to draw any firm conclusions from immunohistochemical studies. What was the post-mortem delay, knowing the sensitivity of the proteins?
2. Fluorescent S100beta immunostainings revealing human astrocytes are not convincing. The problem with astrocyte labeling is that the antibodies available reveal only a limited part of the volume of these cells, which are far more complex and have leaflets invisible to conventional microscopy. This is where most of the relevant (synaptic) mitochondrial labeling is found, and not in the cell bodies.

We thank Reviewer #1 for the comments.

The authors only mention the literature from research groups supporting their own hypothesis, without mention publications demonstrating the existence of oxidative metabolism coming from different substrates in human astrocytes (Eraso-Pichot et al., 2018; Kottinen et al., 2019; Natarajaseenivan et al., 2018; Potter et al., 2019; Zysk et al., 2023, as for example) and murine astrocytes (Azarias et al., 2011; Ioannu et al., 2019; Farmer et al., 2019; Qi et al., 2021 to give few examples) and using a wide range of assays, including enzymatic activity and oxygen consumption. Moreover, in the introduction, authors only cite old data from Bolaños group when their last paper demonstrate fatty acid mitochondrial metabolism (Morant-Ferrando et al., 2023). All these studies need to be taken into account and results presented in the present manuscript need to be compared to the previous literature.

>Response: In the revised manuscript, all of the above literature has been added and extensively discussed. The concept of fatty acid mitochondrial metabolism in astrocytes published by the Bolaños group is particularly emphasized. The possibility of NADH and FADH₂ provision from FAO (in lieu of TCA dehydrogenases) is also now emphasized in the revised text. Specifically, it is now mentioned in page 4 that the group of Bolaños recently demonstrated that adult mouse astrocytes exhibit fatty acid oxidation (FAO) that organizes mitochondrial supercomplexes for sustaining reactive oxygen species formation and cognition¹. The concept of astrocytic mitochondria catabolizing fatty acids has also been demonstrated in murine brain or cultures by the groups of Misgeld², Liu³, Johnson⁴ and Yin⁵. Readouts from other substrates have been examined by other groups specifically for astrocytes, but only from murine origin⁶. Only in the studies by the groups of Malm⁷, Erlandsson⁸ and Langford⁹ were the astrocytes from human origin, but they were either induced pluripotent cell (hiPSC)-derived astrocytes generated from neuroepithelial-like stem cells⁸, or induced pluripotent stem cells of AD patients from skin biopsies or blood samples⁷ or foetal brain tissues that were subject to culturing conditions for several days⁹. FAO generates reducing equivalents in abundance that could be oxidized by the respiratory chain. Here, we have not investigated whether FAO enzymes can be immunocytochemically detected in astrocytes of human brains. The group of Masgrau¹⁰ inferred FAO in human astrocytic mitochondria by performing unbiased statistical comparisons of the mitochondrial transcriptomes in human (and mouse) astrocytes vs neurons using a database compiled by the group of Barres¹¹. However, considering that the above studies were performed in murine tissues or induced human stem cells in cultures, we stress that to the best of our knowledge, it is currently not known whether astrocytes in the adult human brain engage in FAO, and if yes, what is the fate of the reducing equivalents output. What we can also respond to Reviewer #1, is that we queried the transcripts of FAO-related genes (ACAD10, ACADS, ACADM, ECHS1, HADH, CPT1A) in the Allen Brain Atlas. Our analysis reveals that in human brains, non-neuronal cells hardly express these genes and at a much lower levels compared to neuronal cells. This distinction, however, is not present in mouse brains. Please note that this latter information has not been included in the revised manuscript.

In general, methodology is not clearly described not justified. Some concerns on the labelling and quantification methodology are: could the different size of neurons vs astrocytes alter the quantifications as measurements are normalized by area? Importantly, according to methods (lines 230-232), the detection of neuronal and astrocytic borders were performed through mitochondrial markers for the first case and astrocytic markers for the second. The different criteria could clearly alter the measurements. Experiments should be performed in neuronal and astrocytic markers to clearly specify the borders of each cell type.

>Response: The sizes of neurons and astrocytes are indeed different, the neurons are bigger. The way we presented the quantitative data was a density value, which means the number of immunopositive pixels were divided by the average size of the cells. Thus, when the density values are much higher for neurons than for astrocytes, it means that the absolute number of labelled pixels per cell exhibits an even greater marked difference. Reviewer #1 is correct to point out that the different criteria used to identify neurons and astrocytes could alter the measured variables. Therefore, in the revised manuscript, we applied the neuronal marker HuC/HuD to identify neurons and their borders. For verification purposes, the quantification of CoxIV was performed in the dorsomedial prefrontal cortex. The total area of labelled mitochondria expressed in percentage of the total area of the cell was 9.06 ± 0.54 (n=92 cells) by using HuC/HuD to identify neurons, and 9.83 ± 0.84 (n=52 cells) when the neurons were identified using the mitochondrial marker. The very similar values validate the use of mitochondrial markers. All of the above information are included in the revised manuscript and new data shown in supplementary figure 6.

Supplementary datasets of mRNA of OXPHOS should be clearer explained (subjects, exact methodology) and discussed, as they constitute key data.

>Response: In the revised manuscript, mRNA datasets and relevant results and interpretations are much more emphasized. A whole new section in methodology has been added, two new figures (supplementary figures 7 and 8) are now included and Abstract and Results and Discussion sections have been modified to include more of the results from these datasets.

Lines 100-101: authors state that they use a large sample but not characteristics of the subjects are given.

>Response: In the revised manuscript, the characteristics of the tissue donors are explicitly mentioned. Specifically, for immunohistochemistry: brain blocks from a 62 years old female, a 56 years old male individual and 58 year old male were used. The donors had no brain-related disorders. Post-mortem delay prior to obtaining the samples was 12, 4 and 22 hours, respectively. Further information regarding these human brains can be obtained from the Human Brain Tissue Bank of the Semmelweis University (<https://semmelweis.hu/hbtb/>). For imaging mass cytometry: Formalin-fixed paraffin-embedded sections of hippocampus were obtained from 8 donors that exhibited no brain-related disorders (Post-mortem delay: range from 16-66hrs, mean \pm SD= 42.48 ± 16.62). Further information regarding the donors and the brain samples can be obtained from the Newcastle Brain Tissue Resource at <https://nbtr.ncl.ac.uk/>. Regarding the analysis of transcripts deposited in the Allen Institute for Brain Science: samples from 107 brains were made available through tissue donors. Clinical summaries and donor characteristics as well as a description of the criteria for acceptance of use are provided in <https://tinyurl.com/Allenhumanprotocols>.

Lines 100-104: COXIV expression levels are around 4 times lower in astrocytes compared to neurons in CA1 regions and around 9 times in astrocytes. Authors conclude and stress in the abstract that "hippocampal astrocytes express evanescent amounts of mitochondrial proteins". This overinterpretation of data is repeated through all the manuscript. First there is expression of this enzyme in astrocytes. Second, such differences might not be translated to any significant difference in oxidative metabolism between these two cell types.

>Response: The overall tone of the revised manuscript has been tuned down regarding the differences in expression levels of CoxIV and other proteins, and data are given without overt using of adjectives. Please see the extensively revised manuscript in the main text marked with changes.

Quantifications should be provided in figure 5.

>Response: This has been addressed in the revised manuscript as follows: Figure 5 shows double immunolabelling of pyruvate carboxylase and MPC1 with the astrocyte marker S100B in the subiculum. Quantification of the fluorescently labelled subiculum sections is presented in the revised Supplementary figure 4. The percentage of cell bodies covered by pyruvate carboxylase-immunoreactivity was high for astrocytes while neurons were not immunolabelled. In contrast, the percentage of cell bodies covered by MPC1-immunoreactivity was high for neurons while it was barely detectable in astrocytes.

Why the expression of MDH2 is not analyzed in all the brain areas that COXIV was analyzed?

>Response: The distribution of MDH2 looked very similar to that of CoxIV, but with an inferior quality of double labelling. We would like to note that human fluorescent double labelling requires several steps to reduce autofluorescence, therefore, only the very best antibodies could be used for that purpose. Since we did not expect that MDH2 would provide much additional information than CoxIV but the results would be less conclusive because of the diminished quality of labelling, we decided not to perform that double staining.

Lines 140-145: authors state “it is safe to assume that O₂ consumption in the human cerebral cortex should be attributed almost entirely to neuronal mitochondria” based on an expression of MDH2 in some areas and no reported expression of KGDHC. To my point of view this is an overstatement because they do not consider all brain area neither fatty acid oxidation that produces NADH and FADH₂ per se.

>Response: In the revised manuscript, this and all other overstatements have been rewritten and toned down. The concept of FAO producing NADH and FADH₂ per se has been especially emphasized, see above, our response to the first criticism of Reviewer #1.

Lines 153-156: authors sound patronizing and to my point of view they are misinterpreting their own data. They state there is no COXIV in brain astrocytes, whereas they data show that there is: “The percentage of the cell body covered by COXIV-immunolabelled particles was 7.82 ± 0.88 (n=28 cells) and 9.83 ± 0.84 (n=52 cells) for neurons in the two brain areas, respectively. In contrast, these values were 0.32 ± 0.18 (n=18 cells) and 1.14 ± 0.14 (n=52 cells) for astrocytes”. “In the human substantia nigra, we found a considerably higher level of Cox IV immunoreactivity in astrocytes compared to the other brain regions, albeit still much less than in neurons (Fig. 3). The percentage of the cell body covered by COXIV-immunolabelled particles was 37.18 ± 1.24 (n=64 cells) for neurons in the substantia nigra while for astrocytes, the value was 6.34 ± 0.76 (n=59 cells).” All these values indicate protein expression and values of substantia nigra astrocytes are higher than values of CA1 neurons. Other authors have previously reported expression of COXIV in substantia nigra human astrocytes (Chen et al., 2021).

>Response: In the revised manuscript, this and all other overstatements have been rewritten and toned down and misinterpretations have been removed. The findings on substantia nigra astrocytes have been emphasized that they don't exhibit as stark contrast to the neurons as seen for other cortical and hippocampal areas.

The terms of glia and astrocytes are equally used, whereas they are no synonyms.

>Response: This is true, and in the revised manuscript the term 'glia' has only been kept when referring to other investigators' work where they addressed the glia but not specifically astrocytes.

I would like to stress that authors only provide data of expression by immunolabelling of some enzymes, but their conclusions point to the no existence of oxidative metabolism. Despite the disagreements previously reported (there is expression of enzymes in astrocytes), without functional data, authors cannot disclose the existence of oxidative metabolism in human astrocytes. Caution in all their conclusions need to be taken.

>Response: In the revised manuscript, a new section specifically mentioning this limitation of our study is now included. We argue that we do not provide functional evidence of the absence of oxidative metabolism in these cells. It is important to note that demonstrating cell-specific oxidative metabolism within the human brain is currently unfeasible. While cell-specific OXPHOS can be shown in cell cultures or live murine brains, as extensively documented by others and referenced in our revised manuscript, such evidence for astrocytes within the human brain remains elusive. Furthermore, we would like to also stress that the transcriptomics analysis corroborate our data, and they are now more extensively addressed in the revised manuscript.

We thank Reviewer #2 for the comments.

1. Lines 22-28 – Abstract. Whilst the main notion of the work is summarized in this section, this reviewer has some suggestions for amendments aimed to try to accurately obtain the conclusions specifically obtained from this work. The suggested abstract text is as follows: "The astrocyte-to-neuron lactate shuttle model entails that, upon glutamatergic neurotransmission, glycolytically derived pyruvate in astrocytes is mainly converted to lactate instead of being entirely catabolized in mitochondria. The mechanism of this metabolic rewiring and its occurrence in human brain are unclear. Here by using immunohistochemistry and imaging mass cytometry we show that astrocytes of the adult human neocortex and hippocampal formation express evanescent amounts of mitochondrial proteins critical for performing oxidative phosphorylation (OXPHOS). With less OXPHOS, human brain astrocytes are thus bound to produce more lactate to avoid interruption of glycolysis."

>Response: In the revised manuscript, the abstract has been rewritten as Reviewer #2 suggested. We have also added the information that our data are corroborated by queries of transcriptomes of neuronal versus non-neuronal cells fetched from the Allen Institute for Brain Science for genes coding for a much larger repertoire of entities contributing to OXPHOS, showing that human non-neuronal elements barely expressed mRNAs coding for such proteins.

2. In general, throughout the manuscript the authors use strong assertions such as “lack of ...”. For instance, they used “lack of Cox IV”. However, it would be more accurate to read “undetected Cox IV” or a similar statement that does not imply negation. Any negating statement should be avoided given that they rely on the sensitivity and specificity of the techniques used, so there are not absolute assertions. Please, amend the manuscript accordingly.

>Response: In the revised manuscript, we have toned down our statements and removed all strong assertions. Please see manuscript with marked changes.

3. Lines 131 – ref. 43 refers to MCT1 (monocarboxylate transporter-1), not to Mitochondrial Pyruvate Carrier-1 (MPC1) as the authors determined in this manuscript.

>Response: Indeed, thank you for noticing this. It has been addressed in the revised manuscript.

4. Lines 140-141 – “This, together with the fact that human cortical astrocytes do not express KGDHC(29) essentially mitigates any reasoning for having OXPHOS in them.” This sentence is an over-interpretation of the data. The absence of labeling of some TCA cycle enzymes does not directly demonstrate lack of TCA activity and, furthermore, does not imply no need for OXPHOS. There are alternative metabolic sources for NADH and FADH in astrocytes that could feed the OXPHOS such as, e.g., the fatty acids mitochondrial β -oxidation (Morant-Ferrando et al., Nat Metab. 5, 1290-1302, 2023; doi: 10.1038/s42255-023-00835-6.). The authors should mention this possibility.

>Response: In the revised manuscript, all overinterpretations have been removed; the concept of FAO producing NADH and FADH₂ per se has been especially emphasized, please see above, our response to the first criticism of Reviewer #1.

Lines 142-142 – “Therefore, it is safe to assume that O₂ consumption in the human cerebral cortex should be attributed almost entirely to neuronal mitochondria”. Whilst the data provided in this manuscript and the previously published evidence might be compatible with this notion, this assertion is rather speculative and should be tuned down. For instance, the authors could consider stating something like “Therefore, this evidence is compatible with the notion that O₂ consumption in the human cerebral cortex might be mostly attributed to neuronal mitochondria.”.

>Response: In the revised manuscript, this text has been re-written as suggested by Reviewer #2.

6. Lines 157-159 – This paragraph seems to be misplaced as stands. The authors should consider placing this observation somewhere else within the R&D section, but not at the end of the section after the major conclusions were described.

>Response: In the revised manuscript, this paragraph has been re-allocated above, within the same section.

We thank Reviewer #3 for the comments.

The data obtained in this study do not significantly advance knowledge in this field, apart from the fact that it concerns human samples for which it is more difficult to carry out functional studies. Unfortunately, the study of enzyme expression alone (molecular or protein) does not allow us to advance this scientific question in any meaningful way.

>Response: In the revised manuscript, a new section specifically mentioning this limitation of our study is now included. We argue that we do not provide functional evidence of the absence of oxidative metabolism in these cells. It is important to note that demonstrating cell-specific oxidative metabolism within the human brain is currently unfeasible. While cell-specific OXPHOS can be shown in cell cultures or live murine brains, as extensively documented by others and referenced in our revised manuscript, such evidence for astrocytes within the human brain remains elusive.

1. The number of samples seems to me too small to draw any firm conclusions from immunohistochemical studies. What was the post-mortem delay, knowing the sensitivity of the proteins?

>Response: In the revised manuscript, the following information have been added: for immunohistochemistry: brain blocks from a 62 years old female, a 56 years old male individual and 58 year old male were used. Post-mortem delay prior to obtaining the samples was 12, 4 and 22 hours, respectively. For imaging mass cytometry: Formalin-fixed paraffin-embedded sections of hippocampus were obtained from 8 donors that exhibited no brain-related disorders (Post-mortem delay: range from 16-66hrs, mean \pm SD= 42.48 \pm 16.62). Regarding the analysis of transcripts deposited in the Allen Institute for Brain Science: samples from 107 brains were made available through tissue donors. Clinical summaries and donor characteristics (including the post-mortem delay) are provided in <https://tinyurl.com/Allenhumanprotocols>.

2. Fluorescent S100beta immunostainings revealing human astrocytes are not convincing. The problem with astrocyte labeling is that the antibodies available reveal only a limited part of the volume of these cells, which are far more complex and have leaflets invisible to conventional microscopy. This is where most of the relevant (synaptic) mitochondrial labeling is found, and not in the cell bodies.

>Response: We agree with Reviewer #3 that astrocytes can have long processes, some of which may have escaped detection using the astrocytic marker S100B. However, we do not assume that the content of mitochondria in the processes is very different from the content in the cell bodies because there is a continuous turnover of mitochondria within the cell and novel gene expression is not expected in mitochondria far from the nucleus. Furthermore, the imaging mass cytometry data used GFAP which stains better than S100B regarding the astrocytic long processes, yet support the same conclusions, namely that GFAP-positive cells harbor OXPHOS-critical enzymes in cortical and hippocampal areas to a much lower level than in neurons. Nevertheless, we added this concern/limitation outlined by Reviewer #3 in page 3 of the Results and Discussion of the revised manuscript.

REFERENCES CITED

- 1 Morant-Ferrando, B. *et al.* Fatty acid oxidation organizes mitochondrial supercomplexes to sustain astrocytic ROS and cognition. *Nat Metab* **5**, 1290-1302 (2023). <https://doi.org:10.1038/s42255-023-00835-6>
- 2 Fecher, C. *et al.* Cell-type-specific profiling of brain mitochondria reveals functional and molecular diversity. *Nat Neurosci* **22**, 1731-1742 (2019). <https://doi.org:10.1038/s41593-019-0479-z>
- 3 Ioannou, M. S. *et al.* Neuron-Astrocyte Metabolic Coupling Protects against Activity-Induced Fatty Acid Toxicity. *Cell* **177**, 1522-1535 e1514 (2019). <https://doi.org:10.1016/j.cell.2019.04.001>
- 4 Farmer, B. C., Kluemper, J. & Johnson, L. A. Apolipoprotein E4 Alters Astrocyte Fatty Acid Metabolism and Lipid Droplet Formation. *Cells* **8** (2019). <https://doi.org:10.3390/cells8020182>
- 5 Qi, G. *et al.* ApoE4 Impairs Neuron-Astrocyte Coupling of Fatty Acid Metabolism. *Cell Rep* **34**, 108572 (2021). <https://doi.org:10.1016/j.celrep.2020.108572>
- 6 Azarias, G. *et al.* Glutamate transport decreases mitochondrial pH and modulates oxidative metabolism in astrocytes. *J Neurosci* **31**, 3550-3559 (2011). <https://doi.org:10.1523/JNEUROSCI.4378-10.2011>
- 7 Kontinen, H. *et al.* PPARbeta/delta-agonist GW0742 ameliorates dysfunction in fatty acid oxidation in PSEN1DeltaE9 astrocytes. *Glia* **67**, 146-159 (2019). <https://doi.org:10.1002/glia.23534>
- 8 Zysk, M. *et al.* Amyloid-beta accumulation in human astrocytes induces mitochondrial disruption and changed energy metabolism. *J Neuroinflammation* **20**, 43 (2023). <https://doi.org:10.1186/s12974-023-02722-z>
- 9 Natarajaseenivasan, K. *et al.* Astrocytic metabolic switch is a novel etiology for Cocaine and HIV-1 Tat-mediated neurotoxicity. *Cell Death Dis* **9**, 415 (2018). <https://doi.org:10.1038/s41419-018-0422-3>
- 10 Eraso-Pichot, A. *et al.* GSEA of mouse and human mitochondriomes reveals fatty acid oxidation in astrocytes. *Glia* **66**, 1724-1735 (2018). <https://doi.org:10.1002/glia.23330>
- 11 Zhang, Y. *et al.* Purification and Characterization of Progenitor and Mature Human Astrocytes Reveals Transcriptional and Functional Differences with Mouse. *Neuron* **89**, 37-53 (2016). <https://doi.org:10.1016/j.neuron.2015.11.013>

Reviewers' comments:

Reviewer #1 (Remarks to the Author):

The new version of the manuscript by Dobolyi and colleagues has improved considerably. However, after giving details of the samples used, there are still some main concerns:

1.- Authors report that immunichemical data came only from 3 subjects. In particular, for hippocampus and parahippocampal gyrus samples are only from 2 subjects and for the prefrontal cortex and substantia nigra from 1 subject. Moreover, all subjects are over 56 year all, thus not representing the metabolism of human brain in adults but in middle-aged subjects. I would suggest that analyses are immunochemical analyses are performed for at least 3 subjects and taking into account a wider range of age. The difficulty of experiments nor other parallel analysis exclude them to perform a minimum of samples.

Alternatively, authors should state clearly this fact in all the manuscript and discuss they results accordingly. Specially, they should introduce modifications in the title and abstract where they state:

“Here by using immunohistochemistry and imaging mass cytometry we show that astrocytes of the adult human neocortex and hippocampal formation express barely detectable amounts of mitochondrial proteins critical for performing oxidative phosphorylation (OXPHOS). “

2.- Although authors present data from transcriptomes, they only validate with immunocytochemistry the expression of cytochrome c and CoxIV as genes from the oxidative phosphorylation. Moreover, they do not analyse the expression of key TCA enzymes, because MDH2 is not a key enzyme or analysed in all the brain. This is a major limitation of the study and should be clearly stated in the manuscript and in the section of limitations of the study. Moreover, for the trascriptomic data analyses they only include MDH2, IDH2 and IDH3. MDH2 is not a key enzyme of the TCA and IDH2 is a NADPH dependent enzyme not directly linked to TCA. Reason for such selection should be given, specially whereas other key TCA enzymes have been shown to be upregulated in astrocytes compared to neurons.

Reviewer #2 (Remarks to the Author):

The authors have successfully and adequately addressed all comments raised by this reviewer and, accordingly, the manuscript is now improved.

We thank Reviewer #1 for the comments.

Reviewer #1 (Remarks to the Author):

The new version of the manuscript by Dobolyi and colleagues has improved considerably. However, after giving details of the samples used, there are still some main concerns:

1.- Authors report that immunochemical data came only from 3 subjects. In particular, for hippocampus and parahippocampal gyrus samples are only from 2 subjects and for the prefrontal cortex and substantia nigra from 1 subject. Moreover, all subjects are over 56 year old, thus not representing the metabolism of human brain in adults but in middle-aged subjects. I would suggest that analyses are immunochemical analyses are performed for at least 3 subjects and taking into account a wider range of age. The difficulty of experiments nor other parallel analysis exclude them to perform a minimum of samples.

Alternatively, authors should state clearly this fact in all the manuscript and discuss they results accordingly. Specially, they should introduce modifications in the title and abstract where they state:

“Here by using immunohistochemistry and imaging mass cytometry we show that astrocytes of the adult human neocortex and hippocampal formation express barely detectable amounts of mitochondrial proteins critical for performing oxidative phosphorylation (OXPHOS). “

> Response: In the revised manuscript, we have added data from one additional brain donor, thus, regarding immunohistochemistry, n=4. Immunohistochemical data are as follows: Regarding hippocampus and the adjacent portion of the parahippocampal gyrus, brain blocks from a 62 years old female and a 56 years old male individual were used. Immunolabelling of the dorsomedial prefrontal cortex (DMPFC) and the substantia nigra (SN) was performed using samples of a 58 year old male, while brain sections of a 60 years old female were used for immunolabelling of the anterior cingulate cortex. The donors had no brain-related disorders. Post-mortem delay prior to obtaining the samples was 12, 4, 22 and 48 hours, respectively.

We acknowledge that even with the increase in sample size by just one brain, “n” is still small. Thus, per the recommendation of Reviewer #1, we have stated this fact more clearly in the revised manuscript by including modifications to the abstract to mention the number of samples per technique used. We also hope that the number of brains used for imaging mass cytometry (n=8) and transcriptomic analysis (n=107) adds considerable credibility to our work.

Regarding the lack of ‘age-spread’ among the brain donors, considering that all subjects are over 56 years old, thus representing the metabolism of human brain in middle-aged subjects, we emphasize the considerable difficulty in obtaining human brain material suitable for immunohistochemistry, not only due to scarcity but also because: i) the material must be suitable for immunohistochemistry (brain samples for transcriptomics or general proteomics require much less stringent criteria for use) that entails preservation of sections in predefined protectants; and ii) human brains exhibit a very high degree of autofluorescence, which is significantly quenched using Sudan black. After this step, the antibodies used must exhibit a degree of fluorescence that exceeds the remaining autofluorescence following the Sudan black staining. This can only be achieved in a very small fraction of human brain material, and only in combination with certain antibodies.

2.- Although authors present data from transcriptomes, they only validate with immunocytochemistry the expression of cytochrome c and CoxIV as genes from the oxidative phosphorylation. Moreover, they do not analyse the expression of key TCA enzymes, because MDH2 is not a key enzyme or analysed in all the brain. This is a major limitation of the study and should be clearly stated in the manuscript and in the section of limitations of the study. Moreover, for the transcriptomic data analyses they only include MDH2, IDH2 and IDH3. MDH2 is not a key enzyme of the TCA and IDH2 is a NADPH dependent enzyme not directly linked to TCA. Reason for such selection should be given, specially whereas other key TCA enzymes have been shown to be upregulated in astrocytes compared to neurons.

> Response: In the revised manuscript, we have added immunohistochemical data using a suitable antibody directed against IDH3 subunit A, a critical subunit for this NADH-forming dehydrogenase. By including data on IDH3 and considering that KGDHC, MDH2, and some SDH subunits' expression in the human brain has been addressed in *Brain Struct Funct.* 2020 Mar;225(2):639 and this study, respectively, the expression of all NADH- and FADH2-forming TCA dehydrogenases (considered key enzymes) has been evaluated. The expression of ATP- or GTP-forming succinate-CoA ligase (all subunits) has been addressed in *Brain Struct Funct.* 2015 Jan;220(1):135 and *J Bioenerg Biomembr.* 2015 Apr;47(1-2):33. Nevertheless, we reiterate that the expression of NADH-forming dehydrogenases in the TCA cycle is of secondary importance for the present work: in cells where CoxIV and cytochrome c are barely detected, the provision of NADH could not support OXPHOS anyway. This is elaborated in the section regarding the provision of reducing equivalents from other sources, such as potentially fatty acid oxidation. Finally, we acknowledge the limitation that IDH3A immunocytochemistry is not shown as double-labeling with a neuronal or astrocytic marker, as this seemed technically unattainable for this particular IDH3A antibody, which was the only one found to yield specific staining. For these data, IDH3A expression appears predominantly neuronal based on cellular morphology. Transcriptomic and/or immunohistochemical data on TCA enzymes other than MDH2, IDH2, and IDH3, as well as other gene-coding proteins in human brains, are provided in the supplementary material Dataset (homo) and Dataset (mus). This includes remaining SDH subunits, transporters, and all remaining ETC components (complexes I and III), as well as those involved in coenzyme Q manufacturing, Fo-F1 ATP synthase, adenine nucleotide translocase, assembly-assisting subunits, and chaperones. Additional data can be found in our previous publications: *Brain Struct Funct.* 2015 Jan;220(1):135, *J Bioenerg Biomembr.* 2015 Apr;47(1-2):33, and *Brain Struct Funct.* 2020 Mar;225(2):639.

We thank Reviewer #2 for the positive remark.

REVIEWERS' COMMENTS:

Reviewer #1 (Remarks to the Author):

The authors have addressed all comments raised by this reviewer.

Response to Reviewer #1

We thank Reviewer #1 for the positive comments.